# Unbiased Estimates for Multilabel Reductions of Extreme Classification with Missing Labels

## Abstract

This paper considers the missing-labels problem in the extreme multilabel classification (XMC) setting, i.e. a setting with a very large label space. The goal in XMC often is to maximize either precision or recall of the top-ranked predictions, which can be achieved by reducing the multilabel problem into a series of binary (One-vs-All) or multiclass (Pick-all-Labels) problems. Missing labels are a ubiquitous phenomenon in XMC tasks, yet the interaction between missing labels and multilabel reductions has hitherto only been investigated for the case of One-vs-All reduction. In this paper, we close this gap by providing unbiased estimates for general (non-decomposable) multilabel losses, which enables unbiased estimates of the Pick-all-Labels reduction, as well as the normalized reductions which are required for consistency with the recall metric. We show that these estimators suffer from increased variance and may lead to ill-posed optimization problems. To address this issue, we propose to use convex upper bounds which trade off an increase in bias against a strong decrease in variance.

## 1 Introduction

Extreme multilabel classification (XMC) is a machine learning setting in which the goal is to predict a small subset of positive (or relevant) labels for each data instance out of a very large (thousands to millions) set of possible labels. Such problems arise for example when annotating large encyclopedia [7, 28], in fine-grained image classification [9], and next-word prediction [25]. Further applications of XMC are recommendation systems, web-advertising and prediction of related searches [1, 29, 17, 6].

Typical datasets in these scenarios are very large, resulting in possibly billions of (data, label) pairs [4], making it impossible for human annotators to check each pair. Even annotating only a few samples fully in order to generate a clean test set can be prohibitively expensive. Therefore, both the available training- and test-data are likely to contain some errors. Fortunately, in many cases it is possible to constrain the structure of the labeling errors. Consider, for example, the case of tagging documents: Here, we can assume that each label with which the document has been tagged has been deemed relevant by the annotator, and thus is relatively surely a correct label. On the other hand, the annotator cannot possibly check hundreds of thousands of negative labels. This leads to the setting of missing labels investigated in this paper, in which only positive labels are affected by noise (they can go missing), whereas negative labels remain unchanged (no spurious labels). This model has been introduced to the XMC setting by Jain et al. [16], along with estimates for the *propensities*, the chance of a relevant label to be observed. Similar models are using in learning-to-rank[20, 27, 37] and recommendation systems[32, 14, 15]. For a formal definition of the setting we refer the reader to section 3, and for a more thorough discussion of prior works on missing labels and related settings to section 6.

Submitted to 36th Conference on Neural Information Processing Systems (NeurIPS 2022). Do not distribute.

A common strategy for learning XMC classifiers is to reduce the multilabel problem [34] into a series of binary [8, 3, 40] or multiclass [18, 38, 31] problems, which then can be solved using existing techniques. Such *loss reductions* can be shown to be consistent for the tasks of maximizing precision at $k$ or recall at $k$, but never both at the same time [24]. For one of these methods, One-vs-All, adaptation to the missing labels setting has been shown to yield an improvement in propensity-scored precision (an unbiased estimate of precision@k) metrics [30]. The reductions consistent for precision lead to loss functions that can be decomposed into a sum of contributions from each label, which means the results of Natarajan et al. [26] can be applied. In contrast, the reductions consistent for recall contain a normalization term that is the inverse of the total number of true labels. This term is also necessary for calculating the recall metric itself, demonstrating the need for unbiased estimates for true, non-decomposable multilabel loss functions.

**Contributions**   Our contributions are **1)** A mathematical model of the missing labels setting that describes the observed labels as a product of an (unknown) mask variable with the true labels. Crucially, this mask can be chosen to be *independent* of the labels (Theorem 1), enabling simple proofs for our theorems. **2)** The unique unbiased estimate (Theorems 2, 3) for arbitrary multilabel losses, and in particular for the loss functions arising from multilabel reductions. The unbiased estimate of a lower-bounded loss need not be lower-bounded, and even for bounded losses the unbiased estimate leads to an increase in variance. Therefore, we develop **3)** a convex upper-bound (Theorem 4) for losses based on the normalized Pick-all-Labels reduction. In the missing-labels setting, the generalization error is composed of two contributions: the error due to overfitting to the specific, observed noise-pattern, and the error because only a finite sample has been observed. We present empirical evidence **4)** that the former can be much stronger than the latter, and may be reduced by switching to the upper bounds.

In the main paper, we provide shortened proofs that illustrate the key steps. Detailed step-by-step proofs can be found in the appendix.

**Notation**   Random variables will be denoted by capital letters $X, Y, \ldots$, whereas calligraphic letters denote sets and lower case letters their elements, $x \in \mathcal{X}, \ldots$. Vectors will be denoted by bold font, $\mathbf{y} \in \mathcal{Y}$, if we plan to make use of the fact that they can be decomposed into components $y_1, \ldots, y_k$, with $\mathbf{y}_{\neg k}$ denoting the vector of all components except the $k$'th. The letters $f$, $g$, $h$ and $\ell$ are reserved for functions, $i$, $j$, $k$ denote integers, $[k]$ is the set $\{1, \ldots, k\}$. We denote with $\mathcal{X}$ the *data space*, $\mathcal{Y} = \{0, 1\}^l$ the *label space* and $\hat{\mathcal{Y}} = \mathbb{R}^l$ the *prediction space*. A dataset is defined through the three random variables $X \in \mathcal{X}$, $\mathbf{Y} \in \mathcal{Y}$, and $\mathbf{Y}^* \in \mathcal{Y}$, that represent the *data*, *observed label*, and *ground truth label*. We mark quantities pertaining to the unobservable ground-truth with a superscript star and call $(X, \mathbf{Y}^*)$ the *clean data*.

# 2   Multilabel Reductions

In Menon et al. [24], five different reductions for turning the multilabel learning problem into a sum of binary or multiclass problems are presented (cf. appendix). In the following, let $\ell_{\text{BC}} : \{0, 1\} \times \mathbb{R} \longrightarrow \mathbb{R}$ be a binary loss and $\ell_{\text{MC}} : [l] \times \mathbb{R}^l \longrightarrow \mathbb{R}$ be a multiclass loss. Below, we present four of those reductions, and rearrange their loss functions so that a common pattern emerges.

For *one-vs-all* (OVA) reduction, each label is considered independently, meaning that for each instance $l$ binary problems are to be solved. This leads to a loss function

$$\ell_{\text{OVA}}^*(\mathbf{y}^*, \hat{\mathbf{y}}) = \sum_{j=1}^{l} \ell_{\text{BC}}(y_j^*, \hat{y}_j) = \sum_{j=1}^{l} y_j^* \left( \ell_{\text{BC}}(1, \hat{y}_j) - \ell_{\text{BC}}(0, \hat{y}_j) \right) + \ell_{\text{BC}}(0, \hat{y}_j). \tag{1}$$

In contrast, *pick-all-labels* (PAL) considers all the positive labels for each instance and tries to minimize their corresponding multiclass loss, leading to

$$\ell_{\text{PAL}}^*(\mathbf{y}^*, \hat{\mathbf{y}}) = \sum_{j:y_j^*=1} \ell_{\text{MC}}(j, \hat{\mathbf{y}}) = \sum_{j \in [l]} y_j^* \ell_{\text{MC}}(j, \hat{\mathbf{y}}). \tag{2}$$

Both approaches are consistent for precision at $k$. In order to make the reductions consistent for recall instead of precision, the label value needs to be replaced with a normalized label

$$\tilde{y}_j^* := \frac{y_j^*}{\sum_{i=1}^l y_i^*} = \frac{y_j^*}{1 + \sum_{i \neq j}^l y_i^*}, \tag{3}$$

where the expression on the right has the advantage of being well defined even if there are no positives for the sample. This leads to the OVA-N and PAL-N reductions. By moving label-independent parts into functions $f$ and $g_j$, the reductions get a common structure

$$\ell^*(\mathbf{y}^*, \hat{\mathbf{y}}) = f(\hat{\mathbf{y}}) + \sum_{j=1}^l z_j^* g_j(\hat{\mathbf{y}}), \tag{4}$$

where $z_j = \tilde{y}_j^*$ for the normalized reductions and $z_j^* = y_j^*$ otherwise. The functions $f$ and $g_j$ are the same for the normalized and regular reduction (see appendix).

# 3 Unbiased Estimates with Missing Labels

We are interested in noisy labels where the noise is such that labels can only go missing. This is described by the next two definitions, where the first gives a phenomenological characterization of the setting, whereas the second defines the mathematical model used to describe it. For this setting we then develop unbiased estimates for the preceding loss reductions, in the sense that for a given loss $\ell^*$ we are looking for a new loss function $\ell$ such that $\mathbb{E}\big[\ell(\mathbf{Y}, \hat{\mathbf{Y}})\big] = \mathbb{E}\big[\ell^*(\mathbf{Y}^*, \hat{\mathbf{Y}})\big]$.

**Definition 1** (Propensity). The missing-labels setting we described informally in the introduction leads to the following conditions on the $l$ random variables

$$\mathbb{P}\big\{Y_j = 1 \mid Y_j^* = 1, \mathbf{Y}^*_{\neg j}, X\big\} =: p_j(X), \qquad \mathbb{P}\big\{Y_j = 1 \mid Y_j^* = 0, \mathbf{Y}^*_{\neg j}, X\big\} = 0 \tag{5}$$

The value $p_j(x) \in (0, 1]$ is called the *propensity* of the label $j$ at point $x$.

Such propensity models have been used in extreme classification [30, 16, 39], learning-to-rank [20, 27, 37], and recommendation systems [32, 14, 15].

The following proposition guarantees that a fixed-propensity unbiased estimator can be used to construct a instance-dependent unbiased estimator

**Proposition 1.** *Let $f^*(X, Y^*)$ be some function such that for fixed propensity $\mathbf{p}$, an unbiased estimate is given by $f_{\mathbf{p}}$, i.e. $\mathbb{E}[f_{\mathbf{p}}(X, Y)] = \mathbb{E}[f^*(X, Y^*)]$. For instance-dependent propensity $\mathbf{p}(x)$, an unbiased estimator of $f^*$ is given by $f_{\mathbf{p}(X)}$.*

*Proof.* Using the law of total expectation gives

$$\mathbb{E}[f^*(X, Y^*)] = \mathbb{E}[\mathbb{E}[f^*(X, Y^*) \mid X]] = \mathbb{E}\big[\mathbb{E}\big[f_{p(X)}(X, Y^*) \mid X\big]\big] = \mathbb{E}\big[f_{p(X)}(X, Y^*)\big]. \quad \square$$

Therefore, we will supress the dependence of the propensity on the data point in the rest of the paper.

The relation between $\mathbf{Y}^*$ and $\mathbf{Y}$ can be modeled by a set of independent *mask* variables $\mathbf{M}$:

**Theorem 1** (Masking Model). *Assuming $\mathbf{Y}^*$ and $\mathbf{Y}$ follow Definition 1, then then there exists a random variable $\mathbf{M} \in \{0, 1\}^l$ such that $\mathbf{Y} = \mathbf{M} \odot \mathbf{Y}^*$ almost surely and $M_j$ is independent of $(\mathbf{Y}^*, X, \mathbf{M}_{\neg j})$ for all $j \in [l]$. It holds that $\mathbb{E}[M_j] = p_j$.*

This can be seen as a multilabel generalization of the similar statement given in Teisseyre et al. [33]. The independent variables $\mathbf{M}$ provide a convenient framework for proving the results that follow, because the independence allows to factorize expectations containing $\mathbf{M}$.

**Proposition 2** (Unbiased Estimate for Decomposable Reductions). *Assume the setting of Definition 1, with the additional condition that the predictions $\hat{\mathbf{Y}}$ are independent of the missing mask $\mathbf{M}$. Then the unbiased estimate for the loss (4) with $z = y$, denoted by $\ell = \mathfrak{P}(\ell^*)$, is given by*

$$\ell(\mathbf{y}, \hat{\mathbf{y}}) = f(\hat{\mathbf{y}}) + \sum_{j=1}^l \frac{y_j}{p_j} g_j(\hat{\mathbf{y}}). \tag{6}$$

The predictions have to be independent of the locations $\mathbf{M}$ where the labels go missing. This is fulfilled if the predictions $\hat{Y} = h(X, \mathbf{W})$ are the output of a classifier $h$ whose weights $\mathbf{W}$ are independent of $\mathbf{M}$.[1]

For the normalized reductions, it would suffice to find an unbiased estimate of $\tilde{Y}$ in order to apply the same argument as above. However, we are not aware of a derivation for such an estimate that is simpler than the fully generic case presented below.

**Theorem 2** (Unbiased Estimate for Non-Decomposable Loss). *For a generic multilabel loss function $\ell^*$, the unbiased estimate $\ell = \mathfrak{P}(\ell^*)$ under the conditions of Theorem 2 is given by*

$$\ell(\mathbf{y}, \hat{\mathbf{y}}) = \sum_{\mathbf{y}' \preceq \mathbf{y}} \prod_{j:y_j=1} \left( \frac{y_j'(2 - p_j) + p_j - 1}{p_j} \right) \ell^*(\mathbf{y}', \hat{\mathbf{y}}), \tag{7}$$

*where $\mathbf{y}' \preceq \mathbf{y}$ means $\{0, 1\} \ni y_j' \leq y_j$.*

This means that for an instance with $k$ positive labels, we need $2^k$ evaluations of the original loss function in order to calculate the unbiased estimate. This is only feasible because, despite having a very large label space, typical extreme-classification datasets have only few positives per instance.

Unfortunately, the division by (products of) propensity values means that the unbiased estimates will have much larger variance than the original loss function would have on clean data. As an illustrative example, consider the binary case in the limit $p \ll 1$. We can show that in this case the variance grows with $p^{-1}$ compared to the evaluation on clean data.

**Proposition 3** (Increase in Variance). *Setting $q^* := \mathbb{E}[Y^*]$ and $\ell = \mathfrak{P}(\ell^*)$, for small propensities $p \ll 1$, the variance increases with the inverse of the propensity, $\mathbb{V}[\ell(Y, \hat{y})] \approx \frac{1}{p(1-q^*)} \mathbb{V}[\ell^*(Y^*, \hat{y})]$.*

This means that in the binary case the variance increases linearly with inverse propensity. In the multilabel case, this is amplified further due to the product of propensities.

The result above raises the question whether there might be other unbiased estimators with reduced variance. For example, the conditional expectation $\mathbb{E}[\ell^*(Y^*, X)|Y]$ also gives an unbiased estimate with lower variance, but cannot be calculated without knowledge of the conditional probabilities $\mathbb{P}\{Y \mid X\}$. The following theorem states that $\ell = \mathfrak{P}(\ell^*)$ is unique if we want the loss function to work for all possible distributions of data. Thus we cannot reduce the variance.

**Theorem 3** (Uniqueness). *Let $p_j \in (0, 1] \; \forall j \in [l]$. For an arbitrary loss function $\ell^*$, let $\ell$ and $\ell'$ be unbiased versions, in the sense that for all $X, \mathbf{Y}, \mathbf{Y}^*$ that fulfill the masking model Theorem 1 with propensity $\mathbf{p}$, it holds*

$$\mathbb{E}[\ell^*(\mathbf{Y}^*, X)] = \mathbb{E}[\ell(\mathbf{Y}, X)] = \mathbb{E}[\ell'(\mathbf{Y}, X)]. \tag{8}$$

*Then, $\ell' = \ell$.*

The unavoidable increase in variance indicates that there might be a bias-variance trade-off between using the unbiased loss that may overfit more strongly on the observed noise, and using the original loss function which gives wrong results even if $n \to \infty$. If one calculates a standard Rademacher bound for generalization (see appendix), this error bound increases with a factor $\frac{2-p}{p}$.[2]

In a classical learning setup, the generalization error would be described by the difference between the empirical risk and the true risk $\hat{R}_{\ell^*}^*\left[\hat{h}\right] - R_{\ell^*}^*\left[\hat{h}\right]$. However, in the case of missing labels, this can be decomposed in two ways

$$R_{\ell^*}^*[h] - \hat{R}_\ell[h] = \overbrace{R_{\ell^*}^*[h] - R_\ell[h]}^{=0} + R_\ell[h] - \hat{R}_\ell[h] \tag{9}$$

$$= \underbrace{R_{\ell^*}^*[h] - \hat{R}_{\ell^*}^*[h]}_{\text{finite sample}} + \underbrace{\hat{R}_{\ell^*}^*[h] - \hat{R}_\ell[h]}_{\text{noise pattern}}, \tag{10}$$

Whereas the first equation is just a restatement of the unbiasedness, the second contains some new insight: The generalization error can be decomposed into the difference between the true risk $R_{\ell^*}^*[h]$

---

[1] In this sense, we will use the notation $\ell(y, x)$ to evaluate a loss also on a data point.

[2] The bound in this paper corresponds to Natarajan et al. [26, Thm. 9], though that published result is wrong and missing the increase in the bound due to the increased range of the function.

and the empirical risk on clean training data $\hat{R}^*_{\ell*}[h]$, and the difference between that and the estimated empirical risk on observed data $\hat{R}_\ell[h]$. Because the classifier $h$ depends (through $Y = \mathbf{M} \odot Y^*$) on the mask variables, $\ell$ does not give an unbiased estimate (on training data) and thus the second term is non-zero even in expectation. In fact, in the low-regularization regime this term may dominate the entire error, as we will demonstrate in section 5.

# 4 Convex Upper-Bounds

The unbiased estimate allows us to calculate the loss even on data with missing labels, but can we also use it for training? Ideally, the loss function should be lower-bounded, so the minimization is well defined, it should be convex so the minimum is unique. Further, the variance of the unbiased estimator should not be too large, so that a reasonable amount of training samples is sufficient.

If we assume $\ell_{\text{BC}}$ and $\ell_{\text{MC}}$ to be lower-bounded and convex, then only the PAL-reduction results in an unbiased estimate that is guaranteed to have the same properties, as it is a positive combination of $\ell_{\text{MC}}$. Due to the uniqueness result, it is not possible to find an unbiased estimate that is always convex for the other reductions. Thus, in order to make them amenable for training, we propose to switch from unbiased estimates to convex upper-bounds. Below we present solutions for the OvA and normalized PAL-reduction. The normalized OVA-reduction remains an open problem.

**Upper-Bound for OvA-Reduction** The OvA-reduction is based on a binary loss, which often is a convex surrogate for the 0-1 loss. To get a convex loss in the missing-labels case, we thus switch the order of operations [30, 5]: Instead of taking an unbiased estimate of a convex surrogate, we form a convex surrogate of an unbiased estimate. Taking $\theta$ to be a thresholding function (e.g. $\theta(s) = \mathbb{1}\{s > 0\}$), the 0-1-loss can be written as

$$\ell^*_{0-1}(y, \hat{y}) = y\theta(\hat{y}) + (1 - y)(1 - \theta(\hat{y})) \tag{11}$$

with unbiased estimate

$$\ell_{0-1}(y, \hat{y}) = \left(\frac{2}{p_j} - 1\right) y\theta(\hat{y}) + (1 - y)(1 - \theta(\hat{y})) + y\left(\frac{p_j - 1}{p_j}\right). \tag{12}$$

As the last term does not depend on the predictions, it can be dropped for an optimization objective. If $\ell_{\text{BC}}(1, \hat{y})$ is a convex upper-bound on $\theta(\hat{y})$ and $\ell_{\text{BC}}(0, \hat{y})$ on $(1 - \theta(\hat{y}))$, so that overall $\ell_{\text{BC}}$ is a convex upper-bound on the 0-1 loss, then performing these substitutions gives a convex loss function for the OvA-reduction:

$$\tilde{\ell}_{\text{OvA}}(\mathbf{y}, \hat{\mathbf{y}}) = \sum_{j=1}^{l} \left(\frac{2}{p_j} - 1\right) y_j \ell_{\text{BC}}(1, \hat{y}_j) + (1 - y_j)\ell_{\text{BC}}(0, \hat{y}_j) \tag{13}$$

**Upper-Bound for Normalized PAL-Reduction** We have formulated the normalized multilabel reductions in terms of the variable $\tilde{Y}^*$. A naive attempt of correcting for the noisy labels by replacing $Y^*$ with $Y/p$ is not unbiased. However, the resulting estimator $\tilde{Y}$ turns out to be an upper bound. The two estimators are given by

$$\tilde{Y}^*_i = \frac{Y^*_i}{1 + \sum_{j \neq i} Y^*_j}, \qquad \tilde{Y}_i := \frac{Y_i/p_i}{1 + \sum_{j \neq i} Y_j/p_j}. \tag{14}$$

**Theorem 4** (Normalized Label Upper-Bound). *Under the conditions of Theorem 2, replacing the true label with the unbiased estimate of the observed label as shown in Equation 14 results in an upper bound, whose error itself can be bounded by a data-dependent term*

$$\mathbb{E}\left[\tilde{Y}^*_i\right] + \sum_{j \neq i} \left(\frac{1 - p_j}{p_j}\right) \mathbb{E}\left[\frac{Y_i}{p_i} \cdot \frac{Y_j}{p_j}\right] \geq \mathbb{E}\left[\tilde{Y}_i\right] \geq \mathbb{E}\left[\tilde{Y}^*_i\right]. \tag{15}$$

*Proof.* For convenience denote $S^*_i := \sum_{j \neq i} Y^*_j$ and $S_i := \sum_{j \neq i} Y_j/p_j$, and note that $S_i$ is independent of $M_i$. By pulling out known factors and using the independence of $M$ and $\mathbf{Y}^*$ we can show that

$$\mathbb{E}[S_i \mid \mathbf{Y}^*] = \sum_{j \neq i} \mathbb{E}\left[M_j Y^*_j/p_j \mid \mathbf{Y}^*\right] = \sum_{j \neq i} Y^*_j \, \mathbb{E}[M_j/p_j \mid \mathbf{Y}^*] = S^*_i. \tag{16}$$

Expanding terms and using independence of $M_i$, then applying the tower property and pulling out the measurable factor results in

$$\mathbb{E}\left[\tilde{Y}_i\right] = \mathbb{E}\left[\frac{M_i Y_i^*/p_i}{1+S_i}\right] = \mathbb{E}\left[\frac{M_i}{p_i}\right]\mathbb{E}\left[\frac{Y_i^*}{1+S_i}\right] = \mathbb{E}\left[\mathbb{E}\left[\frac{Y_i^*}{1+S_i}\ \middle|\ \mathbf{Y}^*\right]\right] = \mathbb{E}\left[Y_i^*\,\mathbb{E}\left[\frac{1}{1+S_i}\ \middle|\ \mathbf{Y}^*\right]\right].$$

The function $h : \mathbb{R}_{\geq 0} \longrightarrow \mathbb{R}$ given by $t \mapsto 1/(1+x)$ is convex, because its second derivative is $2(1+t)^{-3}$, which is larger than zero for non-negative $t$. Because $S_i \geq 0$ almost surely, we can apply Jensen's inequality to the inner expectation and use (16)

$$\mathbb{E}\left[\tilde{Y}_i\right] \geq \mathbb{E}\left[\frac{Y_i^*}{1+\mathbb{E}[S_i \mid \mathbf{Y}^*]}\right] = \mathbb{E}\left[\frac{Y_i^*}{1+S_i^*}\right] = \mathbb{E}\left[\tilde{Y}_i^*\right].$$

On the other hand, we can use the Taylor formula with intermediate point $\zeta \in [S_i, S_i^*]$ to expand

$$\frac{1}{1+S_i} = \frac{1}{1+S_i^*} - \frac{S_i - S_i^*}{\left(1+S_i^*\right)^2} + \frac{(S_i - S_i^*)^2}{(1+\zeta)^3}. \tag{17}$$

Using $\zeta \geq 0$ to bound the denominator, then multiplying with $Y_i^*$ and taking the expectation gives

$$\mathbb{E}\left[\frac{Y_i^*}{1+S_i}\right] \leq \mathbb{E}\left[\frac{Y_i^*}{1+S_i^*}\right] + \mathbb{E}\left[Y_i^*(S_i - S_i^*)^2\right]. \tag{18}$$

The variance term can be calculated by substituting $S_i$ and $S_i^*$, expanding the sum, and using the independence of $M$ to show that the mixed terms are zero:

$$\mathbb{E}\left[Y_i^*(S_i - S_i^*)^2\right] = \mathbb{E}\left[Y_i^*\left(\sum_{j\neq i} Y_j^*\left(\frac{M_j}{p_j}-1\right)\right)^2\right]$$

$$= \sum_{j\neq i}\mathbb{E}\left[Y_i^*(Y_j^*)^2\left(\frac{M_j}{p_j}-1\right)^2\right] + \sum_{j\neq i}\sum_{k\notin\{i,j\}}\mathbb{E}\left[Y_i^*Y_j^*Y_k^*\right]\mathbb{E}\left[\frac{M_j}{p_j}-1\right]\mathbb{E}\left[\frac{M_k}{p_k}-1\right]$$

$$= \sum_{j\neq i}\mathbb{E}\left[Y_i^*Y_j^*\right]\mathbb{E}\left[\frac{M_j}{p_j^2}-2\frac{M_j}{p_j}+1\right] = \sum_{j\neq i}\left(\frac{1-p_j}{p_j}\right)\mathbb{E}\left[\frac{Y_i}{p_i}\cdot\frac{Y_j}{p_j}\right]. \quad \square \tag{19}$$

Note that the transformation of equation (3) was crucial for this calculation, because it makes the mask variables in the numerator and denominator independent.

In practice, most entries of the co-occurrence matrix $\mathbb{E}[Y_i \cdot Y_j]$ will be extremely small, caus-

Table 1: Error bound for XMC datasets

| Dataset | Average | Worst Case |
|---|---|---|
| Eurlex-4K | 0.02 | 0.51 |
| AmazonCat-13K | 0.0006 | 0.24 |

ing only a minute contribution to the error bound. This can be illustrated by calculating, on two real datasets, the upper-bound for the error of the proposed estimator, by approximating $\mathbb{E}[Y_i \cdot Y_j]$ with the label co-occurrence frequency. The propensities are estimated as in Jain et al. [16]. Looking at the mean value, and the worst case for any label (Table 1), We can see that the error on average is very small, indicating that the worst-case bound only applies to very few labels.

**Corollary 1** (PAL Upper-Bound). *Under the assumptions of Theorem 2, if the underlying multiclass loss $\ell_{MC}$ is a non-negative convex function, the expression*

$$\tilde{\ell}(\mathbf{y},\hat{\mathbf{y}}) := \sum_{j=1}^{l} \frac{y_i/p_i}{1+\sum_{j\neq i}y_j/p_j}\ell_{MC}(j,\hat{\mathbf{y}}) \tag{20}$$

*gives a nonnegative, convex upper-bound on the true normalized PAL loss in expectation.*

## 5 Experimental Results

In this section we present some empirical evidence that illustrates the influence of missing labels and the unbiased estimates and upper bounds on overfitting and bias-variance trade-off. Additional results and a more detailed description of the procedure can be found in the appendix.

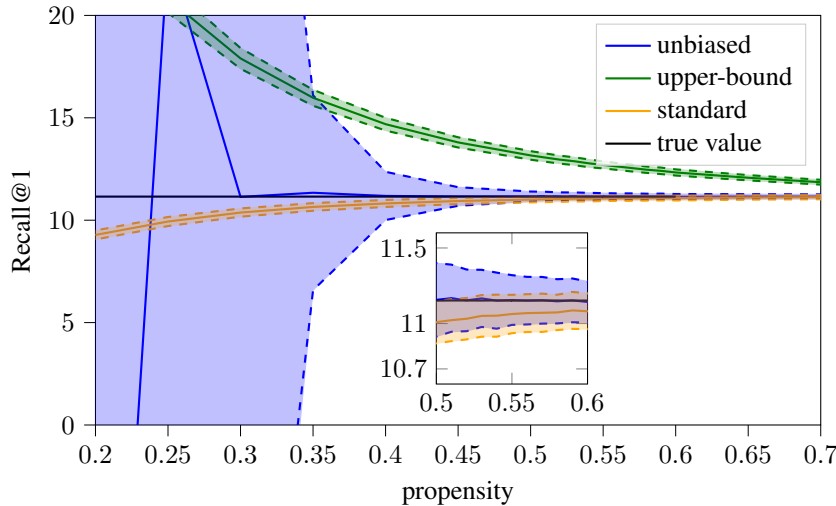

Figure 1: Unbiased estimate of per-example recall with artificial data as described in the main text. The shaded region corresponds to one standard deviation, estimated over 100 repetitions. The black line denotes the true recall.

**Prediction Setting**    First, we want to demonstrate the variance problem in a simple prediction setting, where the classifier is fixed and we want to determine its performance. Consider a setting in which there are 100 different labels, which are independent and each has a probability of 10%. We randomly draw $10\,000$ ground-truth label vectors, and generate observed labels by removing according to a propensity $p$ that is identical for all labels. The predictions are generated by randomly choosing a label from the ground-truth. We calculate the average per-example recall using the standard estimator, the unbiased estimator, and the upper bound, and plot the results in Figure 1.

As can be seen, for moderate propensities the unbiased estimator works well, but for propensities below $0.45$ the $10\,000$ samples are not sufficient to get an accurate estimate. In this setting, the upper-bound results in a larger error than using the standard estimator.

**Training Setting**    Ideally, we would benchmark our loss functions on a real XMC task. However, for those we neither know the exact propensities, nor can we validate that the unbiased estimates and upper bounds produce reasonable results, since the fully-labeled ground truth is unknown.

Instead of using fully artificial data, we chose to construct a dataset based on existing data: We took `AmazonCat-13k`[22] and consider only the 100 most common labels, which are the ones with the highest propensity according to Jain et al. [16]. We artificially remove labels according to inverse propensity, which increases linearly based on the ordering of label frequencies, such that the most common label has an inverse propensity of 2 and the 100th most common one of 20. This process partially preserves the strong imbalances that are typical of extreme classification datasets.

On this data, we train a linear classifier with $L_2$-regularization using different basis loss functions with **a)** the original (standard) loss on clean training data and **b)** noisy training data, as well as **c)** the unbiased version and **d)** the upper-bound version on noisy data. For each training run, we evaluate the loss on noisy and clean training and test data. For the evaluation on noisy data, the corresponding unbiased estimators are used.

In this linear-classifier experiment, the noise-pattern overfitting is much stronger than the overfitting due to finite sampling (10). Figure 2 shows this for the case of the BCE loss in OvA-reduction and CCE loss in normalized PAL reduction. For the classifier trained on clean data (blue), the weights are independent of the noise pattern and thus the dashed and dotted lines coincide in expectation. For the case of OvA reduction using the BCE loss, the training loss gets reduced much further using the unbiased loss function or the upper-bound loss function than using the standard loss. This decrease more than compensates the increase in generalization gap, and as such the minimal test loss is better with these two variants of the loss function. In contrast, in the non-decomposable case, even though

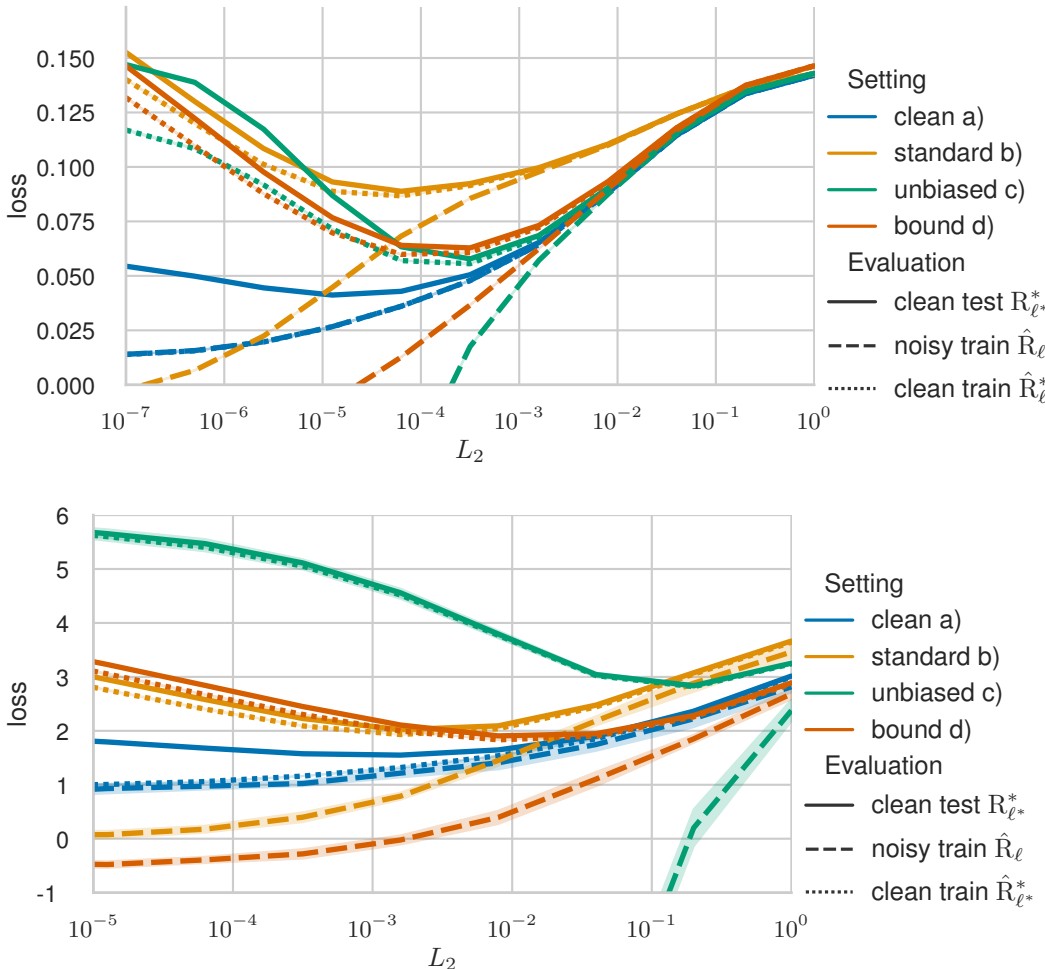

Figure 2: Binary cross-entropy (top) and normalized categorical cross-entropy (bottom) for different regularization strengths, evaluated on noisy training data, clean training data, and clean test data. The gaps between dashed and dotted lines correspond to overfitting to the noise pattern, the smaller gaps between dotted and solid lines show the generalization gaps due to the finite training sample. As the dashed lines are for noisy data, they are calculated using the unbiased estimate (6).

the observed training loss decreases drastically with the unbiased loss, the increase in overfitting makes the test loss worse than using the biased standard loss function.

In this case, using the upper-bound (20) can mitigate the effect, though there is still significant overfitting, as evidenced by the estimated training loss being less than zero. This is possible because even though the loss we use for training is a non-negative upper bound on the expected unbiased loss, the dashed curves show the value estimated for the loss using the unbiased estimator, which can be negative due to overfitting. For the OVA case, the upper bound (13) also reduces overfitting, but does not result in an overall better classifier on test data.

In terms of the bias-variance trade-off, the graphs show a clear trend: The optimal regularization for training on noisy data is larger than on clean data. It is also larger when using the unbiased or upper-bound loss as compared to standard loss. This is as expected from the variance analysis and generalization bound presented in the theory.

## 6 Related Work

**Unbiased Estimates for Noisy Labels** Learning with missing labels is a specific instance of learning with class-conditional noise. For the case of binary labels, unbiased estimates of the loss function can be found in Natarajan et al. [26]. A more general approach is given in Van Rooyen and Williamson [35]. In their notation, $f$ is a function and $\mathbb{P}$ the probability distribution over clean data, that is transformed by the invertible operator $\mathsf{T}$ into a *corrupted* probability distribution. Let $\mathsf{R}$ be the inverse of $\mathsf{T}$, and $\mathsf{R}^*$ its adjoint, then $\langle \mathbb{P}, f \rangle = \langle \mathsf{R} \circ \mathsf{T}(\mathbb{P}), f \rangle = \langle \mathsf{T}(\mathbb{P}), \mathsf{R}^*(f) \rangle$. This equation forms the basis for their "Theorem 5 (Corruption Corrected Loss)", which states that a *corruption corrected* function $l_{\mathrm{R}}$ is given $\forall a \in \mathcal{A}$ by $l_{\mathrm{R}}(\cdot, a) = \mathsf{R}^*(l(\cdot, a))$, where $\mathcal{A}$ denotes the set of possible actions that will be evaluated by the loss functions. For a finite label space with $n$ possible, the operator $\mathsf{R}^*$ can be represented with an $n \times n$ matrix. For the multilabel case here, applying this naively would require $2^l$ evaluations of the original loss function. In contrast, the direct approach presented in section 3 is much more efficient.

**Alternatives** In some settings with noisy labels, it is possible to use a learning algorithm that is inherently noise tolerant [12, 36]. Certain performance objectives such as the balanced error or the AUC are noise robust even under the more general setting of mutually contaminated distributions as shown in Menon et al. [23]. A data re-calibration approach tries to identify from the training data which samples are corrupted, e.g. by looking at samples for which the network is very unsure, and adapt the training process correspondingly [13, 42, 19] It is also possible to first train a scorer on the noisy data naively, from which a classifier adapted to a given rate of missing labels can be constructed by choosing an appropriate threshold [23]. Similarly, the inference procedure of PLTs can be adapted to take into account a propensity model [39].

**Related Learning Settings** Learning with missing labels is highly related to learning from positive and unlabeled (PU) data [11]. An unbiased loss function for this setting is given in Du Plessis et al. [10]. The appearing difficulties, that non-negativity and convexity need not be preserved, are the same as in our setting [21]. A slightly different setting with missing labels is given by semi-supervised learning, where it is know for which labels are missing [41].

## 7 Summary and Discussion

This paper provides unbiased estimates for four cases of multilabel reductions given in Menon et al. [24]. Except for the PAL reduction, these estimators can be non-convex and even negatively unbounded. The unbiased estimates come with an increase in variance. This is unavoidable if unbiasedness is required, as the estimators can be shown to be unique. If sufficient training data is available, then the unbiased loss functions can be used, but for the normalized reductions we found that even 1.2 million instances in AmazonCat are not enough. Much fewer data points are needed in order to estimate the overall loss of a classifier. This is because for training, an accurate estimate for $\mathbb{E}[\ell(Y^*, h(X) \mid X]$ needs to be formed, whereas for evaluation this is averaged over the entire dataset, $\mathbb{E}[\ell(Y^*, h(X)]$. This indicates that the unbiased estimates can be useful for hyperparameter tuning and model selection.

For training, however, another approach is needed. A method that fixes the negative unboundedness and non-convexity and also reduces the variance is to switch to a convex upper-bound. We have shown that this can stabilize the training and improve the results.

Furthermore, the data in section 5 suggest training with missing labels requires more regularization, irrespective of whether training uses standard-, unbiased-, or convex upper-bound losses. Our findings agree with Arpit et al. [2] who found that typical regularizers prevent a deep network from memorizing *noisy* examples, while not hindering the learning of patterns from *clean* instances.

All in all, our results show that a) unbiasedness can be achieved for generic multilabel losses, and in particular the losses resulting from multilabel reduction, but also that b) these losses might not be suitable for optimization. We have presented one method that trades away unbiasedness for the ability to handle training with lower amounts of data. An exciting future research prospect would be to investigate families of loss functions that can continuously trade off bias and variance, and thus allow for optimal training with different amounts of available data.

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
