# OpenReview forum: "Unbiased Estimates for Multilabel Reductions of Extreme Classification with Missing Labels"
_NeurIPS.cc/2022/Conference — NeurIPS 2022 Submitted_

### Official Review · Reviewer_s9K4 · 2022-07-11

**Rating:** 5
**Confidence:** 3
**Soundness:** 3 good
**Presentation:** 1 poor
**Contribution:** 3 good

**Summary:**

The paper studies the problem of extreme multi-label learning with missing labels. To solve the problem, the paper considers reducing the multi-label problem into a serious binary or multi-class problems. To deal with missing labels, the paper proposes unbiased estimates for multi-label reduction loss functions. To reduce the variance of estimates, the paper proposes convex upper-bound for the OVA and PAL-reduction.

**Questions:**

In experiments, why use BCE for OVA reduction and CCE for PAL reduction?

In Related Work, authors claim that “learning with missing labels is a specific” instance of learning with class-conditional noise. Why? Is it there any reference?

**Ethics Review Area:**

["I don’t know"]

**Limitations:**

The experiments are weak due to the lack of comparison with counterpart methods. The results on real-world scenarios are also insufficient.

**Strengths And Weaknesses:**

Strength

1.The paper proposes a framework for dealing with missing labels in extreme multi-label setting. In the proposed framework, the unbiased estimates in terms of decomposable reduction and non-decomposable losses under missing-label setting are derived.

2.To deal with issue of increased variance, the paper proposes convex upper bounds for normalized PAL reduction, which provides trade-off between bias and variance.

Weakness

1.The experiments are weak. The paper only conducts a toy experiment and an experiment on a real-world dataset. Furthermore, the proposed method does not compare with other counterpart methods. The experimental results on the current submission are insufficient to validate the effectiveness of the proposed method.

2.The paper is not well written. Many important details are missed or placed on the appendix, which make the paper hard to understand. For example, Eq.(4) is difficult to understand without any detailed introduction (in appendix). This makes this paper become difficult to follow. The organization of this paper should be improved. For example, the figures in experiments are too large with limited information.

3.The paper fails to sufficiently discuss its pioneering works. Many related works are missed. For example, authors do not discuss the relevant works in extreme multi-label classification that are very relevant to this work. The very relevant works about multi-label noise are also missed.

---

> ### Author Response · Authors · 2022-08-02
> **Authors Response**
>
> 1. **OVA and PAL reductions** : By the nature of the reductions, OVA needs at its foundation a binary loss function, and PAL needs a multiclass loss function. Categorical cross-entropy (CCE) with softmax link function is the de-facto default loss for multiclass classification. In the binary case, there is more variety, e.g. hinge loss is another popular loss in XMC, but since we used cross-entropy for the PAL case, we decided to go with cross-entropy (with logistic link function) also for the OVA setting.
>
> 2. **Class-conditional noise** :For the binary/decomposable case, class-conditional noise (i.e. $P[Y=0|Y*=1] = \rho_+$,$P[Y=1|Y*=0] = \rho_-$; the probability of having an error depends on the label) with one side $\rho_-$ of the noise set to zero corresponds to missing labels. This class-conditional noise is discussed in the [Natarajan] paper cited in the sentence immediately following this statement.
> For the true, non-decomposable multilabel case, it might have been better to state that noisy labels are a specific instance of corrupted labels (see the [Van Rooyen] paper also mentioned in this paragraph). Learning with corrupted labels has a general transition matrix $P[\mathbf{Y} | \mathbf{Y}^*]$, learning with missing labels imposes the condition that the probability be zero for all transitions in which the observed labels are not a subset of the true labels.
>
> 3. **Further experiments** : We have added experiments with additional datasets in appendix B.4. However, note the caveat that the propensity values used there are only crude estimates, leading to values outside of the [0\% to 100\%] range. It may be noted that we are not proposing any new method in this paper, so attempting to draw algorithmic conclusions such as by the comment "experimental results on the current submission are insufficient to validate the effectiveness of the proposed method." is not the right perspective for this paper.
>
> 4. **Other related works on XMC with missing labels** : Could you give an example of a very relevant related paper that we have missed?

---

### Official Review · Reviewer_6sB5 · 2022-07-11

**Rating:** 5
**Confidence:** 4
**Soundness:** 3 good
**Presentation:** 2 fair
**Contribution:** 2 fair

**Summary:**

This paper discusses the problem of  missing labels in extreme multi-label classification (XMC). First a framework for defining the loss function is defined, which can be applied to various one-vs-all (OVA) and pick-all-labels reduction methods. Next, unbiased estimators for the loss functions based on propensity  are discussed, and it is shown that such estimators have high variance. Next, convex upper-bounds for the loss functions are discussed, in order to obtain efficient solvers. Numerical results on two datasets are presented to illustrate the various aspects of missing labels in XMC problems.


**Questions:**

Missing labels is an important problem in XMC literature. The paper might be presenting an interesting framework for solving XMC problems with missing labels.
However, the current version of the paper has the following shortcomings:

1. Significance: Unbiased loss functions for XMC with missing labels have been proposed before, see [1],[2]. It is not clear how the proposed estimators in this paper differ from the existing ones and what are the significance of the proposed estimators. Also, certain aspects are  not clear, eg., why the upper bounds would have reduced variance and how the bias looks like in these cases.

2. Numerical results: The experiment results seem limited. Only two datasets are considered. Many aspects of the framework are not studied, e.g., whether the considers loss functions solve the original XMC problem efficiently, what type of XMC algorithms perform best (OvA or PAL), also, comparison to [1],[2] would be helpful to better understand the advantages of the popped framework.

3. Scalibity: Since in XMC set up, the number of instances and labels can be very large (in millions), it is not clear if OvA or PAL is scalable. There are many other reduction methods (label reduction or tree based methods) that are scalable. So, it is not clear if studying OvA type methods is useful in XMC settings.


4. Minor comment:
i. In Theorem 2, under the conditions of Theorem 2 --> under the conditions of Proposition 2?


References:

[1] Schultheis, Erik, and Rohit Babbar. "Unbiased Loss Functions for Multilabel Classification with Missing Labels." arXiv preprint arXiv:2109.11282 (2021).

[2] Schultheis, Erik, et al. "Unbiased Loss Functions for Extreme Classification With Missing Labels." arXiv preprint arXiv:2007.00237 (2020).


**Limitations:**

Yes, the paper addresses these.

**Strengths And Weaknesses:**

The strengths of the paper  are:
1. An important problem in XMC, that of missing labels is discussed.
2. Unbiased loss functions and their convex upper bounds are discussed.
3. The framework is general and addressed a range of XMC methods that can be written as OvA or PAL.

The weaknesses of the paper  are:
1. Significance of the results are not clear.
2. Numerical results seem limited.
3. Scalability issues are not addressed.

---

> ### Author Response · Authors · 2022-08-02
> **Authors response**
>
> ### Variance of Upper Bound
> An intuitive explanation why the upper bound results in reduced variance is that, in the computation of the unbiased estimate, we divide by the *product* of several propensity, and thus by potentially very small numbers. In contrast, the bound only divides by single propensities.
>
> ### Significance
> Note that [2] is just a preprint version of [this paper](https://dl.acm.org/doi/10.1145/3442381.3450139), which has already been cited and discussed in our submitted paper.
> That paper is solely focused on the decomposable case, whereas we are trying here to provide the theoretical foundations to treat the non-decomposable case. Correspondingly, we have marked eq. (6) as a Proposition, whereas the main result eq. (7) is designated as a Theorem.
>
> ### Results
> We have added experiments with additional datasets in appendix B.4. However, note the caveat that the propensity values used there are only crude estimates, leading to values outside of the [0\% to 100\%] range.
>
> ### Scalability
> Note that recent methods such as XR-Transformer, while they do use a tree based structure for feature learning, achieve their best results when combined in a vary large linear OvA model together with tf-idf features of the input, so OvA is still a useful concept in XMC. Note that PAL reductions can be made scalable, e.g. in hierarchical softmax.
>
>
> ### Other
> Proposition 2: Yes, thanks.

---

### Official Review · Reviewer_YLGN · 2022-07-13

**Rating:** 6
**Confidence:** 3
**Soundness:** 3 good
**Presentation:** 3 good
**Contribution:** 3 good

**Summary:**

In this paper, the authors address the problem of unbiased estimates for multi-label reductions of extreme classification with missing labels. Specifically, the unbiased estimates for four cases of multi-label reductions are discussed. The authors first show the unbiased estimates for general multi-label losses capable for pick-all-labels and normalized reductions, and then prove the upper-bounds for two reductions, which trade-off bias and variance. The empirical experiments demonstrate the influence of missing labels, as well as unbiased estimates. The results also indicate that although unbiased loss functions work well with sufficient training data, normalized reductions can still be starving. Besides, the authors also show that generic multi-label losses with achievable unbiasedness could not be suitable for optimization while the presented trade-off method works well for handling training with fewer data.

**Questions:**

* Most of the existing state-of-the-art methods for XMC problems are tree-based methods. In each layer, not all of the labels will participate in optimization and loss calculations. Moreover, the predictions and inference also usually are based on beam search. I wonder if the conclusions and guidelines can be also applied to those methods.

* The authors model missing-label setting by treating them as independent noises, but real-world noises can be more complicated with many different distributions and correlations among labels. I wonder if the methods can be further extended to more different types of noises (or distributions of missing labels).

**Limitations:**

The authors did not address any limitations and potential negative societal impact.

**Strengths And Weaknesses:**

Strengths
* Comprehensive coverage on different types and cases reductions.
* Theorems and proofs for both unbiased estimates and convex upper-bounds.
* Empirical studies and practical conclusions that emphasize the contributions

Weaknesses
* Lack of discussions on the relations to tree-based XMC methods, which are one of the most popular conventional methods for tackling XMC problems.
* Lack of discussions of different types of noises.

---

> ### Author Response · Authors · 2022-08-02
> **Authors Response**
>
> Recent tree-based techniques aimed towards scalability are algorithmic constructs, while the focus of our work is on the theoretical aspects of learning with missing labels. In principle, it could be applied at each level of the tree, though this requires defining a suitable propensity for the meta-labels. Such as attempt has been undertaken in the [recent paper](https://dl.acm.org/doi/10.1145/3442381.3450139), where propensities of the meta-labels are computed using the number of instances under the meta-label node in the label tree.
> Note that, once a model is trained with a loss function adapted to missing labels, the inference procedure need not be adapted. This is in contrast to probabilistic label trees, where you can keep the training procedure unchanged, but then have to adapt the inference to the missing labels as described in [this paper](https://dl.acm.org/doi/abs/10.1145/3404835.3463084).
>
> Regarding your question on the model of noise: As stated in our Proposition 1, the theory can be extended to instance-dependent noise without problems. However, in practice the difficulty would be in having a model of $p_j(x)$. In such a case, this also means that the labels go missing independently from each other *conditional* on the features, so this assumption is reduced to conditional independence.
> Finally, in the decomposable case, since the loss function does not allow for any (nonlinear) interaction between the labels, the condition that labels go missing independently from each other can be lifted, and Proposition 2 would still hold.

---

### Official Review · Reviewer_uLF6 · 2022-07-14

**Rating:** 5
**Confidence:** 4
**Soundness:** 3 good
**Presentation:** 3 good
**Contribution:** 3 good

**Summary:**

This paper tackles the missing label problem in the Extreme Classification setting. In particular -
1) It provides unbiased estimation of all kind of losses (decomposable or non decomposable over labels) under the missing label setting. Previous papers mostly focused on decomposable losses.
2) Shows that if propensities are low then these losses can have very high variance and may lead to ill-posed optimization problems.
3) Proposes to use convex upper bounds of these losses which significantly reduces the variance at cost of slight increase in bias.


**Questions:**

It would have been nice to see if the proposed convex upper bound framework actually improves the performance on standard tasks/datasets. I believe for a lot of public datasets the propensity scores have been provided [1] so it should be pretty simple to try them out and compare against methods which do not take variance reduction into account.

[1] The Extreme Classification Repository: Multi-label Datasets & Code: http://manikvarma.org/downloads/XC/XMLRepository.html

**Limitations:**

Yes

**Strengths And Weaknesses:**

Originality: While the issue of high variance with inverse propensity models has been explored previously such as in [1], this paper studies it in the extreme classification setting where it hasn't been explored much.

Quality: In terms of experimental results I feel like the paper is a bit weak. It would have been good to see some comparison of the proposed technique with other popular variance reduction methods such as one proposed in [1]


[1] Swaminathan, Adith, and Thorsten Joachims. "Counterfactual risk minimization: Learning from logged bandit feedback." International Conference on Machine Learning. PMLR, 2015

---

> ### Author Response · Authors · 2022-08-03
> **Authors Response**
>
> ### Variance Reduction
> As far as we can see, what  [Adith, Joachims] provides is more a model selection procedure
> that takes variance into account, instead of a training method that could directly be plugged into the XMC framework. This is because estimating the variance during the optimization run is rather non-trivial, and requires adaptations like in Proposition 1 of that paper.
>
> A second form of variance reduction, also employed in that paper, though already proposed earlier, is clipping of the propensities. We can add such experiments to the paper. Note, however, that this still can be problematic in the non-decomposable case, because the inverse propensities are multiplied together there, and even with clipped propensities this could cause large factors to arise. Additionally, to keep the large sum in our equation (7) tractable, our implementation uses random subsampling of its summands, which acts as a further source of variance. In a preliminary experiment clipping propensities at 0.9 for Eurlex data,
> we found that this could stabilize the training (in contrast to wild fluctuations observed in unbiased training), but did not result in improvements over the vanilla loss function.
>
> ### Results on XMC repository datasets
> Even though it is straightforward to train models using our loss functions on the (smaller) datasets from
> the extreme classification repository, the interpretation of the results in quite unclear.
> The problem is that the propensity values that are provided there (following the [Jain](http://manikvarma.org/pubs/jain16.pdf) paper), at best, very crude estimates.
> This leads the unbiased estimates to result in values outside the range of [0\%, 100\%].
> This problem is typically
> hidden in current practice in XMC by defining propensity-scored precision/nDCG to include an additional
> normalization step that forces the results to be in the 0\% - 100\% range. For an in-depth discussion about
> the problems of current usage of propensity models in XMC, we refer to [this paper](http://arxiv.org/abs/2207.13186).

---

> > ### Comment · Reviewer_uLF6 · 2022-08-08
> > **Response to rebuttal**
> >
> > Thanks for your response. I agree with your comment that its hard to interpret results as the unbiased estimates do not fall in [0 -100] and just normalising them to fall in between 0-100 is probably not the best idea. However, we can still do some sort of proxy/qualitative analysis where we check if the model trained with unbiased loss is actually giving better results for tail labels (as done in Jain et al. paper as well), because thats where most of the missing data would be.
> > But anyways, I am willing to increase my rating to 6.

---

### Author Response · Authors · 2022-08-02
**Rebuttal**

We thank the reviewers for their comments. Please find below our response to the points raised by individual reviewers.

---

### Author Response · Authors · 2022-08-03
**Aim of the paper**

We want to use this opportunity to clarify that the aim of this paper is not to present a single algorithm that gets a new SOTA result
on some benchmark dataset (which, given the uncertainty around the propensity values that are currently available, would be difficult to
judge in any case), but instead provide a broad foundation that offers recipies how to deal with missing labels for a wide variety (i.e. using different kinds of loss reduction) of current and potential future algorithms:


Currently, in the field of XMC many new publications acknowledge the missing-labels problem
by providing PSP metrics as their evaluation, but do not adapt their training procedure. We think this is because all
those publications follow the results of [Jain] that provide a very limited set of the tools, which we significantly extend
leading to a more complete picture. Therefore, we want to provide a broad set of foundational tools (i.e. covering
multiple reductions), as e.g. different losses and decompositions work best for different datasets (see Table 4 in the appendix, where
normalized PAL is best, vs Table 5 where BCE is best). PAL and OVA also offer different opportunities for algorithms, e.g. hierarchical softmax approaches correspond to PAL reduction.

We further want to provide an analysis of these tools, which is difficult to
achieve with the real-world datasets as their propensities are only rough guesses, making the interpretation of the results
difficult. For this reason, we have performed the in-depth experiment using a modifified version of real data, where we have control over
the propensity values.

Finally, having the tools to use propensity during training might also help
spur further research into the refining propensity model, which has essentially remained unchanged since its publication.
Overall, the paper is meant to give additional, non-trivial, insights into the problem of missing labels in XMC. We do
not aim at introducing new algorithms beating the state-of-the-art in terms of empirical results.

---

### Meta-Review · Area_Chair_zJat · 2022-08-30

**Recommendation:** Reject
**Confidence:** Less certain

**Metareview:**

This is a borderline paper. The reviewers lean towards acceptance but do have reservation about the paper. I have also read the paper in order to make an informed recommendation. In the end I've decided to recommend against acceptance at this point for the reasons I'll describe below. I do understand, and sympathize with the authors' frustration at my decision, but I truly believe that the paper can be significantly improved with some additional work.

The authors tackle an interesting and important problem in the area of extreme classification (and multi-label classification in general): the problem of missing labels due to various biases in how the data is annotated. This is a well recognized and important problem both in the extreme classification literature, and also in the benchmark creation literature, especially for computer vision tasks. (I recommend the authors look at the work on Imagenet 2.0 and similar papers where due to the shortcoming of the annotation protocol relevant labels are missing).  The paper makes an interesting theoretical contribution on how to obtain unbiased loss estimates in this case, identifies an issue with the high variance of these estimates, and proposed a lower-variance upper-bound that could be used as a surrogate loss.

The main drawback of the paper, as originally submitted is the very limited experimental evaluation. The authors explain the lack of empirical results by the fact that there are no datasets that have accurate propensity scores available, so even evaluating on such datasets would be meaningless. While it is true that existing datasets do have this problem, it is incumbent on the authors to find an application where their work would be applicable. Otherwise, if there is no application that can benefit from this work, what is it useful for?  One suggestion I can make to do the following evaluation: take a (sub-sample) of the instances  that method M assigned label L to and, using human labelers, determine how many of those documents should truly be assigned label L and how many should not have label L. Do this for multiple labels L.  While I admit that this would be a tedious endeavor, it would be possible to achieve with current crowdsourcing technology, and it would significantly improve the paper.

During the author response period, the authors have submitted additional results on some real datasets. These results do look pretty strong, but, because they have been rushed and not really integrated in the paper, I fear is difficult for the reviewers to truly scrutinize their validity, and to draw the correct conclusions from them.  The paper would benefit from having the authors integrate the results in the paper and fully analyze them.

I do believe this is interesting work, and I encourage the authors to revise and resubmit their paper at a future conference. But as it is, the paper is not yet ready for publication.

**Award:**

No

---

### Decision · Program_Chairs · 2022-09-14

Reject